# ADVERSARIAL INDUCTIVE TRANSFER LEARNING WITH INPUT AND OUTPUT SPACE ADAPTATION

## ABSTRACT

We propose Adversarial Inductive Transfer Learning (AITL), a method for addressing discrepancies in input and output spaces between source and target domains. AITL utilizes adversarial domain adaptation and multi-task learning to address these discrepancies. Our motivating application is pharmacogenomics where the goal is to predict drug response in patients using their genomic information. The challenge is that clinical data (i.e. patients) with drug response outcome is very limited, creating a need for transfer learning to bridge the gap between large pre-clinical pharmacogenomics datasets (e.g. cancer cell lines) and clinical datasets. Discrepancies exist between 1) the genomic data of pre-clinical and clinical datasets (the input space), and 2) the different measures of the drug response (the output space). To the best of our knowledge, AITL is the first adversarial inductive transfer learning method to address both input and output discrepancies. Experimental results indicate that AITL outperforms state-of-the-art pharmacogenomics and transfer learning baselines and may guide precision oncology more accurately.

## 1 INTRODUCTION

Deep neural networks (Goodfellow et al., 2016) have demonstrated the state-of-the-art performance in different problems, ranging from computer vision and natural language processing to genomics (Eraslan et al., 2019) and medicine (Topol, 2019). However, these networks often require a large number of samples for training, which is challenging and sometimes impossible to obtain in the real world applications.

Transfer learning (Pan & Yang, 2009) attempts to solve this challenge by leveraging the knowledge in a *source* domain, a large data-rich dataset, to improve the generalization performance on a small *target* domain. Training a model on the source domain and testing it on the target domain violates the i.i.d assumption that the train and test data are from the same distribution. The discrepancy in the input space decreases the prediction accuracy on the test data, which leads to poor generalization (Zhang et al., 2019). Many methods have been proposed to minimize the discrepancy between the source and the target domains using different metrics such as Jensen Shannon Divergence (Ganin & Lempitsky, 2014), Maximum Mean Discrepancy (Gretton et al., 2012), and Margin Disparity Discrepancy (Zhang et al., 2019). While transductive transfer learning (e.g. domain adaptation) uses a labeled source domain to improve generalization on an unlabeled target domain, inductive transfer learning (e.g. few-shot learning) uses a labeled source domain to improve the generalization on a labeled target domain where label spaces are different in the source and the target domains (Pan & Yang, 2009).

Adversarial domain adaptation has shown great performance in addressing the discrepancy in the input space for different applications (Schoenauer-Sebag et al., 2019; Hosseini-Asl et al., 2018; Pinheiro, 2018; Zou et al., 2018; Tsai et al., 2018; Long et al., 2018; Chen et al., 2017; Tzeng et al., 2017), however, adversarial adaptation to address the discrepancies in both the input and output spaces has not yet been explored. Our motivating application is pharmacogenomics (Smirnov et al., 2017) where the goal is to predict response to a cancer drug given the genomic data (e.g. gene expression). Since clinical datasets in pharmacogenomics (patients) are small and hard to obtain, many studies have focused on large pre-clinical pharmacogenomics datasets such as cancer cell lines as a proxy to patients (Barretina et al., 2012; Iorio et al., 2016). A majority of the current methods are trained on cell line datasets and then tested on other cell line or patient datasets (Sharifi-Noghabi

et al., 2019b; Geeleher et al., 2014). However, cell lines and patients data, even with the same set of genes, do not have identical distributions due to the lack of an immune system and the tumor microenvironment in cell lines (Mourragui et al., 2019). Moreover, in cell lines, the response is often measured by the drug concentration that reduces viability by 50% (IC50), whereas in patients, it is often based on changes in the size of the tumor and measured by metrics such as response evaluation criteria in solid tumors (RECIST) (Schwartz et al., 2016). This means that drug response prediction is a regression problem in cell lines but a classification problem in patients. Therefore, discrepancies exist in both the input and output spaces in pharmacogenomics datasets. Table A1 provides the definition of these biological terms.

In this paper, we propose Adversarial Inductive Transfer Learning (AITL), the first adversarial method of inductive transfer learning. Different from existing methods for transfer learning, AITL adapts not only the input space but also the output space. Our motivating application is transfer learning for pharmacogenomics datasets. In our driving application, the source domain is the gene expression data obtained from the cell lines and the target domain is the gene expression data obtained from patients. Both domains have the same set of genes (i.e., raw feature representation). Discrepancies exist between the gene expression data in the input space, and the measure of the drug response in the output space. AITL learns features for the source and target samples and uses these features as input for a multi-task subnetwork to predict drug response for both the source and the target samples. The output space discrepancy is addressed by the multi-task subnetwork, which has one shared layer and separate classification and regression towers, and assigns binary labels (called cross-domain labels) to the source samples. The multi-task subnetwork also alleviates the problem of small sample size in the target domain by sharing the first layer with the source domain. To address the discrepancy in the input space, AITL performs adversarial domain adaptation. The goal is that features learned for the source samples should be domain-invariant and similar enough to the features learned for the target samples to fool a global discriminator that receives samples from both domains. Moreover, with the cross-domain binary labels available for the source samples, AITL further regularizes the learned features by class-wise discriminators. A class-wise discriminator receives source and target samples from the same class label and should not be able to predict the domain accurately.

We evaluated the performance of AITL and state-of-the-art inductive and adversarial transductive transfer learning baselines on pharmacogenimcs datasets in terms of the Area Under the Receiver Operating Characteristic curve (AUROC) and the Area Under the Precision-Recall curve (AUPR). In our experiments, AITL achieved a substantial improvement compared to the baselines, demonstrating the potential of transfer learning for drug response prediction, a crucial task of precision oncology.

## 2 RELATED WORK

### 2.1 TRANSFER LEARNING

Following the notation of (Pan & Yang, 2009), a domain like $DM$ is defined by a raw input feature space[1] $\mathbf{X}$ and a probability distribution $p(X)$, where $X = \{x_1, x_2, ..., x_n\}$ and $x_i$ is the $i$-th raw feature vector of $X$. A task $T$ is associated with $DM = \{\mathbf{X}, p(X)\}$, where $T = \{\mathbf{Y}, F(.)\}$ is defined by a label space $\mathbf{Y}$ and a predictive function $F(.)$ which is learned from training data of the form $(X, Y)$, where $X \in \mathbf{X}$ and $Y \in \mathbf{Y}$. A source domain is defined as $DM_S = \{(x_{s_1}, y_{s_1}), (x_{s_2}, y_{s_2}), ..., (x_{s_{n_S}}, y_{s_{n_S}})\}$ and a target domain is defined as $DM_T = \{(x_{t_1}, y_{t_1}), (x_{t_2}, y_{t_2}), ..., (x_{t_{n_T}}, y_{t_{n_T}})\}$, where $x_s \in X_S, x_t \in X_T, y_s \in Y_S$, and $y_t \in Y_T$. Since $n_T << n_S$ and it is challenging to train a model only on the target domain, transfer learning aims to improve the generalization on a target task $T_T$ using the knowledge in $DM_S$ and $DM_T$ and their corresponding tasks $T_S$ and $T_T$. Transfer learning can be categorized into three categories: 1) unsupervised transfer learning, 2) transductive transfer learning, and 3) inductive transfer learning. In unsupervised transfer learning, there is no label in the source and target domains. In transductive transfer learning, source domain is labeled but target domain is unlabeled, domains can be either the same or different (domain adaptation), but source and target tasks are the same. In inductive transfer learning, target domain is labeled and source domain can be either labeled or unlabeled, and

---

[1]This is different from learned features by the network

domains can be the same or different, but in this category tasks are always different (Pan & Yang, 2009).

## 2.2 INDUCTIVE TRANSFER LEARNING

There are three approaches to inductive transfer learning: 1) deep metric learning, 2) few-shot learning, and 3) weight transfer (Scott et al., 2018). Deep metric learning methods are independent of the number of samples in each class of the target domain, denoted by $k$, meaning that they work for small and large $k$ values. Few-shot learning methods focus on small $k$ ($k \leq 20$). Finally, weight transfer methods require a large k ($k \geq 100$ or $k \geq 1000$) (Scott et al., 2018). Figure A1 (in Appendix) presents this taxonomy.

In drug response prediction, the target domain is small, which means a limited number of samples for each class is available, therefore, few-shot learning is more suitable for such a problem. Few-shot learning involves training a classifier to recognize new classes, provided only a small number of examples from each of these new classes in the training data (Snell et al., 2017). Various methods have been proposed for few-shot learning (Chen et al., 2019; Scott et al., 2018; Snell et al., 2017; Vinyals et al., 2016). For example, Prototypical Networks (ProtoNet) (Snell et al., 2017) constructs prototypical representatives (class means) from source domain learned features and compares the Euclidean distance between the target domain learned features and these class representatives to assign labels to the target samples.

## 2.3 ADVERSARIAL TRANSFER LEARNING

Recent advances in adversarial learning leverage deep neural networks to learn transferable representation that disentangles domain-invariant and class-invariant features from different domains and matches them properly (Peng et al., 2019; Zhang et al., 2019; Long et al., 2018). In this section, we first introduce the Generative Adversarial Networks (GANs) (Goodfellow et al., 2014), and then introduce some of the existing works on adversarial transfer learning.

### 2.3.1 GENERATIVE ADVERSARIAL NETWORKS

GANs (Goodfellow et al., 2014) attempt to learn the distribution of the input $data$ via a minimax framework where two networks are competing: a discriminator $D$ and a generator $G$. The generator tries to create fake samples from a randomly sampled latent variable that fool the discriminator, while the discriminator tries to catch these fake samples and discriminate them from the real ones. Therefore, the generator wants to minimize its error, while the discriminator wants to maximize its accuracy:

$$\underset{G}{Min}\underset{D}{Max}V(G,D) = \sum_{x \sim data} log[D(x)] + \sum_{z \sim noise} log[1 - D(G(z))] \tag{1}$$

A majority of literature on adversarial transfer learning are for transductive transfer learning where the source domain is labeled while the target domain is unlabeled.

### 2.3.2 ADVERSARIAL TRANSDUCTIVE TRANSFER LEARNING

Transductive transfer learning, often referred to as domain adaptation, is the most common scenario in transfer learning. Various methods have been proposed for adversarial transductive transfer learning in different applications such as image segmentation (Chen et al., 2017; Tsai et al., 2018), image classification (Tzeng et al., 2017; Long et al., 2018), speech recognition (Hosseini-Asl et al., 2018), domain adaptation under label-shift (Azizzadenesheli et al., 2019), partial domain adaptation (You et al., 2019), and multiple domain adaptation (Schoenauer-Sebag et al., 2019). The idea of these methods is that features extracted from source and target samples should be similar enough to fool a global discriminator (Tzeng et al., 2017) and/or class-wise discriminators (Chen et al., 2017).

## 2.4 DRUG RESPONSE PREDICTION

The goal of precision oncology is to tailor a treatment for a cancer patient using genomic information of that patient. However, currently, only about 5% of the patients can benefit from precision

oncology because response to a drug is a highly complex phenotype and it depends on diverse genetic and/or non-genetic factors (Marquart et al., 2018).

Pre-clinical pharmacogenomics datasets such as cancer cell lines (Iorio et al., 2016) and patient-derived xenografts (PDX) (Gao et al., 2015) are reliable proxies to study the associations between the genomic landscape and the response to a cancer treatment. The advantage of these resources is that they can be screened with hundreds of drugs – chemotherapy agents and targeted therapeutics – which is impossible for patients. For example, in the Genomics of Drug Sensitivity in Cancer (GDSC) dataset (Iorio et al., 2016) over 1000 pan-cancer cell lines screened with 265 chemotherapy and targeted drugs are available. Another advantage of the pre-clinical datasets is that they are often significantly larger than patient datasets with known drug response (labels).

These advantages of pre-clinical datasets make them a suitable resource to develop computational methods for drug response prediction (Smirnov et al., 2017). Various methods have been developed to predict drug response from single or multiple types of genomic data. For example, Geeleher et al. (2014) proposed a ridge-regression method to predict drug response based on gene expression data. Sharifi-Noghabi et al. (2019b) showed that integrating multiple data types with deep neural networks and transfer learning via sample transfer improves the accuracy of drug response prediction.

## 3    ADVERSARIAL INDUCTIVE TRANSFER LEARNING

### 3.1    PROBLEM DEFINITION

Given a labeled source domain $DM_S$ with a learning task $T_S$ and a labeled target domain $DM_T$ with a learning task $T_T$, where $T_T \neq T_S$, and $p(X_T) \neq p(X_S)$, where $X_S, X_T \in \mathbf{X}$, we assume that the source and the target domains are not the same due to different probability distributions. The goal of Adversarial Inductive Transfer Learning (AITL) is to utilize the source and target domains and their tasks in order to improve the learning of $F_T(.)$ on $DM_T$.

In the area of pharmacogenomics, the source domain is the gene expression data obtained from the cell lines, and the source task is to predict the drug response in the form of IC50 values. The target domain consists of gene expression data obtained from patients, and the target task is to predict drug response in a different form – often change in the size of tumor after receiving the drug. In this setting, $p(X_T) \neq p(X_S)$ because cell lines are different from patients even with the same set of genes. Additionally, $Y_T \neq Y_S$ because for the target task $Y_T \in \{0, 1\}$, drug response in patients is a binary outcome, but for the source task $Y_S \in \mathbb{R}^+$, drug response in cell lines is a continuous outcome. As a result, AITL needs to address these discrepancies in the input and output spaces.

### 3.2    THE AITL METHOD

Our proposed AITL method takes input data from the source and target domains, and achieves the following three objectives: first, it makes predictions for the target domain using both of the input domains and their corresponding tasks, second, it addresses the discrepancy in the output space between the source and target tasks, and third, it addresses the discrepancy in the input space. AITL is a neural network consisting of four components:

- The feature extractor receives the input data from the source and target domains and extracts salient features, which are then sent to the multi-task subnetwork component.

- The multi-task subnetwork takes the extracted features of source and target samples and maps them to their corresponding labels and makes predictions for them. This component has a shared layer and two task-specific towers for regression (source task) and classification (target task). Therefore, by training the multi-task subnetwork on the source and target samples, it addresses the small sample size challenge in the target domain. In addition, it also addresses the discrepancy in the output space by assigning cross-domain labels (binary labels in this case) to the source samples (for which only continuous labels are available) using its classification tower.

- The global discriminator receives extracted features of source and target samples and predicts if an input sample is from the source or the target domain. To address the discrepancy in the input space, these features should be domain-invariant so that the global discriminator cannot predict their domain labels accurately. This goal is achieved by adversarial learning.

- The class-wise discriminators further reduce the discrepancy in the input space by adversarial learning at the level of the different classes, i.e., extracted features of source and target samples from the same class go to the discriminator for that class and this discriminator should not be able to predict if an input sample from a given class is from the source or the target domain.

The AITL cost function consists of a classification loss, a regression loss, and global and class-wise discriminator adversarial losses and is optimized end-to-end. An overview of the proposed method is presented in figure 1.

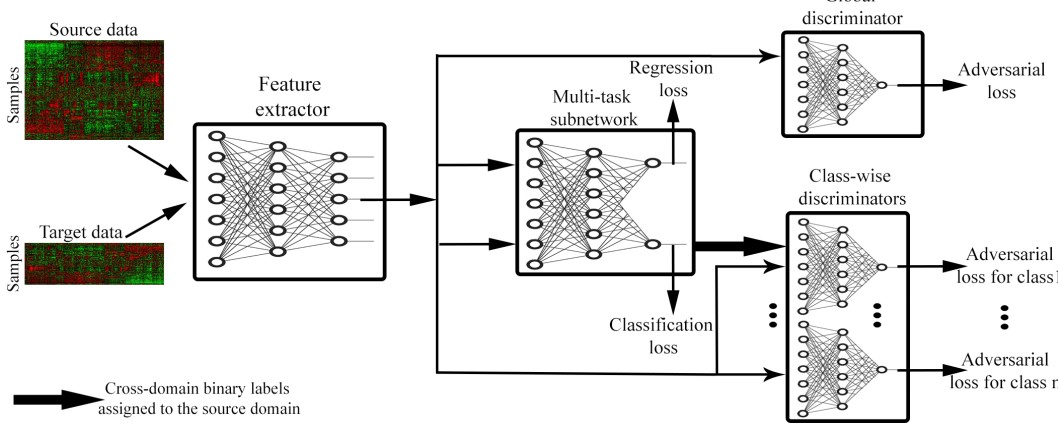

Figure 1: Overview of AITL: First, the feature extractor receives source and target samples and learns feature for them. Then, the multi-task subnetwork uses these features to make predictions for the source and target samples and also assigns cross-domain labels to the source samples. The multi-task subnetwork addresses the discrepancy in the output space. Finally, to address the input space discrepancy, global and class-wise discriminators receive the extracted features and regularize the feature extractor to learn domain-invariant features.

### 3.2.1 FEATURE EXTRACTOR

To learn salient features in lower dimensions for the input data, we design a feature extractor component. The feature extractor is a one-layer fully-connected subnetwork with batch normalization and the ReLU activation function that receives both the source and target samples as input. We denote the feature extractor as $f(.)$:

$$Z_i = f(X_i), i \in \{S, T\} \tag{2}$$

where $Z$ denotes the extracted features for input $X$ which is from either the source $(S)$ or the target $(T)$ domain. In our driving application, the feature extractor learns features for the cell line and patient data.

### 3.2.2 MULTI-TASK SUBNETWORK

After extracting features of the input samples, we want to use these learned features to 1) make predictions for target samples, and 2) address the discrepancy between the source and the target domains in the output space. To achieve these goals, a multi-task subnetwork with a shared layer $g(.)$ and two task-specific towers $M_S(.)$ and $M_T(.)$ is designed, where $M_S$ is for regression (the source task) and $M_T$ is for classification (the target task):

$$\overline{Y_i} = M_i(g(Z_i)), i \in \{S, T\} \tag{3}$$

The performance of the multi-task subnetwork component is evaluated based on a binary-cross entropy loss for the classification task on the target samples and a mean squared loss for the regression task on the source samples:

$$L_{BCE}(X_T, Y_T, f, g, M_T) = - \sum_{(x_t, y_t) \sim (X_T, Y_T)} [y_t \log \overline{y_t} + (1 - y_t) \log(1 - \overline{y_t})] \tag{4}$$

$$L_{MSE}(X_S, Y_S, f, g, M_S) = \sum_{(x_s, y_s) \sim (X_S, Y_S)} (\overline{y_s} - y_s)^2 \tag{5}$$

Where $Y_S$ and $Y_T$ are the true labels of the source and the target samples, respectively, and $L_{BCE}$ and $L_{MSE}$ are the corresponding losses for the target and the source domains, respectively. The multi-task subnetwork component outputs 1) the predicted labels for the target samples, and 2) the assigned cross-domain labels for the source samples. The classification tower in the multi-task subnetwork makes predictions for the source samples and assigns binary labels (responder or non-responder) because such labels do not exist for the source samples. Therefore, the multi-task subnetwork adapts the output space of the source and the target domains by assigning cross-domain labels to the source domain. The multi-task subnetwork has a shared fully-connected layer with batch normalization and the ReLU activation function. The regression tower has two layers with batch normalization and the ReLU activation function. The classification tower also has two fully connected layer with batch normalization and the ReLU activation function in the first layer and the Sigmoid activation function in the second layer. In our driving application the multi-task subnetwork predicts IC50 values for the cell lines and the binary response outcome for the patients. Moreover, it also assigns binary labels to the cell lines which is similar to those of the patients.

### 3.2.3 GLOBAL DISCRIMINATOR

The goal of this component is to address the discrepancy in the input space by adversarial learning of domain-invariant features. To achieve this goal, a discriminator receives source and target extracted features from the feature extractor and classifies them into their corresponding domain. The feature extractor should learn domain-invariant features to fool the global discriminator. In our driving application the global discriminator should not be able to recognize if the extracted features of a sample are from a cell line or a patient. This discriminator is a one-layer subnetwork with the Sigmoid activation function denoted by $D_G(.)$. The adversarial loss for $D_G(.)$ is as follows:

$$L_{advD_G}(X_S, X_T, D_G) = -\sum_{x_s \sim X_S} [\log D_G(f(x_s))] - \sum_{x_t \sim X_T} [\log(1 - D_G(f(x_t)))] \tag{6}$$

### 3.2.4 CLASS-WISE DISCRIMINATORS

With cross-domain binary labels available for the source domain, AITL further reduces the discrepancy between the input domains via class-wise discriminators. The goal is to learn domain-invariant features with respect to specific class labels such that they fool corresponding class-wise discriminators. Therefore, extracted features of the target samples in class $i$, and those of the source domain which the multi-task subnetwork assigned to class $i$, will go to the discriminator for class $i$. We denote such a class-wise discriminator as $DC_i$. The adversarial loss for $DC_i$ is as follows:

$$L_{advDC_i}(X_S, Y_S, X_T, Y_T, DC_i) = -\sum_{(x_s, y_s) \sim (X_S, Y_S)} [\log DC_i(f(x_s))] - \sum_{(x_t, y_t) \sim (X_T, Y_T)} [\log(1 - DC_i(f(x_t)))] \tag{7}$$

In our driving application the class-wise discriminator for the responder samples should not be able to recognize if the extracted features of a responder sample are from a cell line or a patient (similarly for a non-responder sample). Similar to the global discriminator, class-wise discriminators are also one-layer fully-connected subnetworks with the Sigmoid activation function.

### 3.2.5 COST FUNCTION

To optimize the entire network in an end-to-end fashion, we design the cost function as follows:

$$J = L_{BCE} + L_{MSE} + \lambda_G L_{advD_G} + \lambda_{DC} \sum_i L_{advDC_i} \tag{8}$$

Where, $\lambda_G$ and $\lambda_{DC}$ are adversarial regularization coefficients for the global and class-wise discriminators, respectively.

## 4 EXPERIMENTAL RESULTS

### 4.1 DATASETS

In our experiments, we used the following datasets (See Table A2 in the Appendix for more detail):

- The Genomics of Drug Sensitivity in Cancer (GDSC) cell lines dataset, consisting of a thousand cell lines from different cancer types, screened with 265 targeted and chemotherapy drugs. (Iorio et al., 2016)
- The Patient-Derived Xenograft (PDX) Encyclopedia dataset, consisting of more than 300 PDX samples for different cancer types, screened with 34 targeted and chemotherapy drugs. (Gao et al., 2015)
- The Cancer Genome Atlas (TCGA) (Weinstein et al., 2013) containing a total number of 117 patients with diverse cancer types, treated with Cisplatin, Docetaxel, or Paclitaxel.
- Patient datasets from nine clinical trial cohorts containing a total number of 491 patients with diverse cancer types, treated with Bortezomib (Amin et al., 2014; Mulligan et al., 2007), Cisplatin (Silver et al., 2016; Marchion et al., 2011), Docetaxel (Hatzis et al., 2011; Lehmann et al., 2011; Chang et al., 2005), or Paclitaxel (Hatzis et al., 2011; Bauer et al., 2010; Ahmed et al., 2007).

The GDSC dataset was used as the source domain, and all the other datasets were used as the target domain. For the GDSC dataset, raw gene expression data were downloaded from ArrayExpress (E-MTAB-3610) and release 7.0 of the dataset was used to obtain the response outcome. Gene expression data of TCGA patients were downloaded from the Firehose Broad GDAC and the response outcome was obtained from (Ding et al., 2016). Patient datasets from clinical trials were obtained from the Gene Expression Omnibus (GEO), and the PDX dataset was obtained from the supplementary material of (Gao et al., 2015). For each drug, we selected those patient datasets that applied a comparable measure of the drug response. For preprocessing, the same procedure was adopted as described in the supplementary material of (Sharifi-Noghabi et al., 2019b) for the raw gene expression data (normalized and z-score transformed) and the drug response data. After the preprocessing, source and target domains had the same number of genes.

### 4.2 EXPERIMENTAL DESIGN

We designed our experiments to answer the following three questions:

1. Does AITL outperform baselines that are trained only on cell lines and then evaluated on patients (without transfer learning)? To answer this question, we compared AITL against (Geeleher et al., 2014) and (Sharifi-Noghabi et al., 2019b) (MOLI) which are state-of-the-art methods of drug response prediction that do not perform domain adaptation.

2. Does AITL outperform baselines that adopt adversarial transductive transfer learning (without adaptation of the output space)? To answer this question, we compared AITL against (Tzeng et al., 2017) (ADDA) and (Chen et al., 2017), state-of-the-art methods of adversarial transductive transfer learning with global and class-wise discriminators, respectively.

3. Does AITL outperform a baseline for inductive transfer learning? To answer this last question, we compared AITL against (Snell et al., 2017) (ProtoNet) which is the state-of-the-art inductive transfer learning method for small numbers of examples per class.

Based on the availability of patient/PDX datasets for a drug, we experimented with four different drugs: Bortezomib, Cisplatin, Docetaxel, and Paclitaxel. It is important to note that these drugs have different mechanisms and are being prescribed for different cancers. For example, Docetaxel is a chemotherapy drug mostly known for treating breast cancer patients (Chang et al., 2005), while Bortezomib is a targeted drug mostly used for multiple myeloma patients (Amin et al., 2014). Therefore, the datasets we have selected cover different types of anti-cancer drugs.

In addition to the experimental comparison against published methods, we also performed an ablation study to investigate the impact of the different AITL components separately. $AITL-AD$ denotes a version of AITL without the adversarial adaptation components, which means the network only contains the multi-task subnetwork. $AITL-D_G$ denotes a version of AITL without the

Table 1: Performance of AITL and the baselines in terms of prediction AUROC

| Method/Drug | Bortezomib | Cisplatin | Docetaxel | Paclitaxel |
|---|---|---|---|---|
| (Geeleher et al., 2014) | 0.48 | 0.58 | 0.55 | 0.53 |
| MOLI (Sharifi-Noghabi et al., 2019b) | 0.57 | 0.54 | 0.54 | 0.53 |
| (Chen et al., 2017) | 0.54±0.07 | 0.60±0.14 | 0.52±0.02 | 0.58±0.04 |
| ADDA (Tzeng et al., 2017) | 0.51±0.06 | 0.56±0.06 | 0.48±0.06 | did not converge |
| ProtoNet (Snell et al., 2017) | 0.49±0.01 | 0.40±0.003 | 0.40±0.01 | did not converge |
| AITL$-AD$ | 0.69±0.03 | 0.57±0.03 | 0.57±0.05 | 0.58±0.01 |
| AITL$-D_G$ | 0.69±0.04 | 0.62±0.1 | 0.48±0.03 | **0.62±0.02** |
| AITL$-DC$ | 0.69±0.03 | 0.54±0.1 | 0.59±0.07 | 0.59±0.03 |
| AITL | **0.74±0.02** | **0.66±0.02** | **0.64±0.04** | 0.61±0.04 |

global discriminator, which means the network only employs the multi-task subnetwork and class-wise discriminators. AITL$-DC$ denotes a version of AITL without the class-wise discriminators, which means the network only contains the multi-task subnetwork and the global discriminator.

All of the baselines were trained on the same data, tested on patients/PDX for these drugs, and eventually compared to AITL in terms of prediction AUROC and AUPR. Since the majority of the studied baselines cannot use the continuous IC50 values in the source domain, binarized IC50 labels provided by (Iorio et al., 2016) using the Waterfall approach (Barretina et al., 2012) were used to train them. Finally, for the minimax optimization, a gradient reversal layer was employed by AITL and the adversarial baselines (Ganin et al., 2016).

We performed 3-fold cross validation in the experiments to tune the hyper-parameters of AITL and the baselines based on the AUROC. Two folds of the source samples were used for training and the third fold for validation, similarly, two folds of the target samples were used for training and validation, and the third one for the test. The hyper-parameters tuned for AITL were the number of nodes in the hidden layers, learning rates, mini-batch size, weight decay coefficient, the dropout rate, number of epochs, and the regularization coefficients. We considered different ranges for each hyper-parameter and the final selected hyper-parameter settings for each drug and each method are provided in Section A.2 in the Appendix. Finally, each network was re-trained on the selected settings using the train and validation data together for each drug. We used Adagrad for optimizing the parameters of AITL and the baselines (Duchi et al., 2011) implemented in the PyTorch framework, except for the method of (Geeleher et al., 2014) which was implemented in R. We used the author's implementations for the method of (Geeleher et al., 2014), MOLI, and ProtoNet. For ADDA, we used an existing implementation from `https://github.com/jvanvugt/pytorch-domain-adaptation`, and we implemented the method of (Chen et al., 2017) from scratch.

### 4.3 RESULTS

Tables 1 and A3 (Appendix) and Figure 2 report the performance of AITL and the baselines in terms of AUROC and AUPR, respectively. To answer the first experimental question, AITL was compared to the baselines which do not use any adaptation (neither the input nor the output space), i.e. the method of (Geeleher et al., 2014) and MOLI (Sharifi-Noghabi et al., 2019b), and AITL demonstrated a better performance in both AUROC and AUPR for all of the studied drugs. This indicates that addressing the discrepancies in the input and output spaces leads to better performance compared to training a model on the source domain and testing it on the target domain. To answer the second experimental question, AITL was compared to state-of-the-art methods of adversarial transductive transfer learning, i.e. ADDA (Tzeng et al., 2017) and the method of (Chen et al., 2017), which address the discrepancy only in the input space. AITL achieved significantly better performance in AUROC for all of the drugs and for three out of four drugs in AUPR (the results of (Chen et al., 2017) for Cisplatin were very competitive with AITL). This indicates that addressing the discrepancies in the both spaces outperforms addressing only the input space discrepancy. Finally, to answer the last experimental question, AITL was compared to ProtoNet (Snell et al., 2017) – a representative of inductive transfer learning with input space adaptation via few-shot learning. AITL outperformed this method in all of the metrics for all of the drugs.

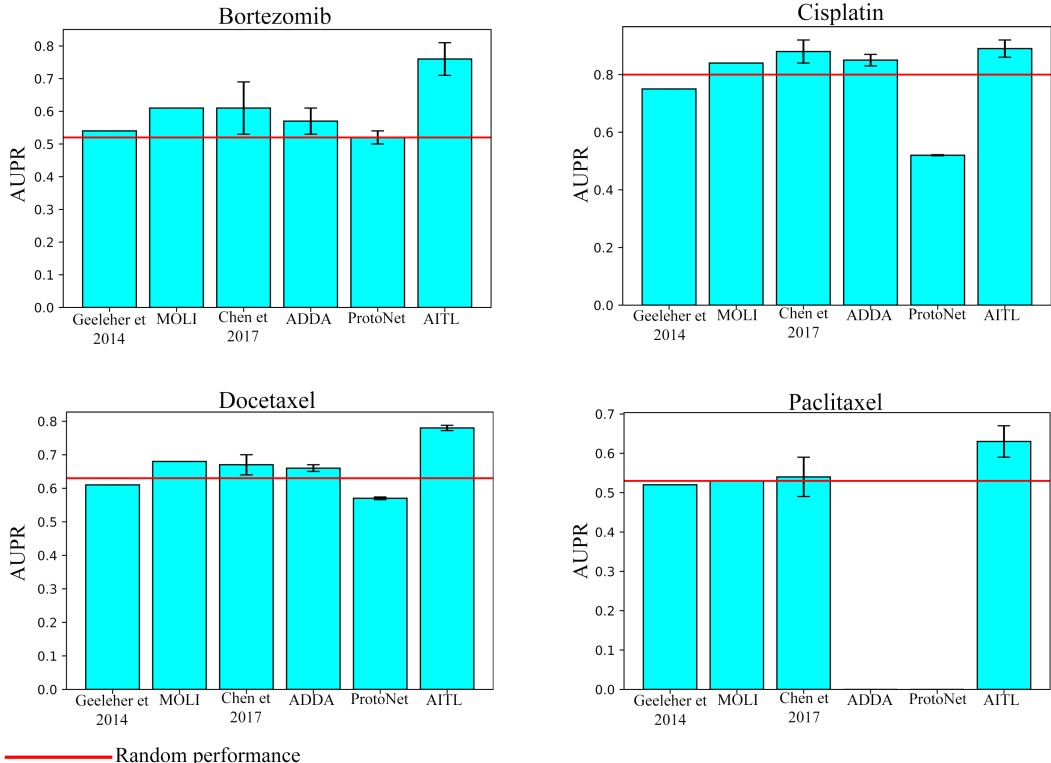

Figure 2: Performance of AITL and the baselines in terms of prediction AUPR

We note that the methods of drug response prediction without adaptation, namely the method of (Geeleher et al., 2014) and MOLI, outperformed the method of inductive transfer learning based on few-shot learning (ProtoNet). Moreover, these two methods also showed a very competitive performance compared to the methods of adversarial transductive transfer learning (ADDA and the method of (Chen et al., 2017)). For Paclitaxel, ADDA did not converge in the first step (training a classifier on the source domain), which was also observed in another study (Sharifi-Noghabi et al., 2019b). ProtoNet also did not converge for this drug.

We observed that AITL, using all of its components together, outperforms all the additional baselines omitting some of the components. This indicates the importance of both input and output space adaptation. The only exception was for the drug Paclitaxel, where AITL$-D_G$ outperforms AITL. We believe the reason for this is that this drug has the most heterogeneous target domain (see Table A1 in the appendix), and therefore the global discriminator component of AITL causes a minor decrease in the performance.

All these results indicate that addressing the discrepancies in the input and output spaces between the source and target domains, via the AITL method, leads to a better prediction performance.

## 4.4 DISCUSSION

To our surprise, ProtoNet and ADDA could not outperform the method of (Geeleher et al., 2014) and MOLI baselines. For ProtoNet, this may be due to the depth of the backbone network. A recent study has shown that a deeper backbone improves ProtoNet performance drastically in image classification Chen et al. (2019). However, in pharmacogenomics, employing a deep backbone is not realistic because of the much smaller sample size compared to an image classification application. Another limitation for ProtoNet is the imbalanced number of training examples in different classes in pharmacogenomics datasets. Specifically, the number of examples per class in the training episodes is limited to the number of samples of the minority class as ProtoNet requires the same number of examples from each class. For ADDA, this lower performance may be due to the lack of end-to-end training of the classifier along with the global discriminator of this method. The reason

is that end-to-end training of the classifier along with the discriminators improved the performance of the second adversarial baseline (Chen et al., 2017) in AUROC and AUPR compared to ADDA. Moreover, the method of (Chen et al., 2017) also showed a relatively better performance in AUPR compared to the method of (Geeleher et al., 2014) and MOLI.

In pharmacogenomics, patient datasets are small or not publicly available due to privacy and/or data sharing issues. We believe including more patient samples and more drugs will increase generalization capability. In addition, recent studies in pharmacogenomics have shown that using multiple genomic data types (known as multi-omics in genomics) works better than using only gene expression (Sharifi-Noghabi et al., 2019b). In this work, we did not consider such data due to the lack of patient samples with multi-omics and drug response data publicly available; however, in principle, AITL also works with such data. Last but not least, we used pharmacogenomics as our motivating application for this new problem of transfer learning, but we believe that AITL can also be employed in other applications. For example, in slow progressing cancers such as prostate cancer, large patient datasets with gene expression and short-term clinical data (source domain) are available, however, patient datasets with long-term clinical data (target domain) are small. AITL may be beneficial to learn a model to predict these long-term clinical labels using the source domain and its short-term clinical labels (Sharifi-Noghabi et al., 2019a). Moreover, AITL can also be applied to the diagnosis of rare cancers with a small sample size. Gene expression data of prevalent cancers with a large sample size, such as breast cancer, may be beneficial to learn a model to diagnose these rare cancers.

## 5 CONCLUSION

In this paper, we introduced a new problem in transfer learning motivated by applications in pharmacogenomics. Unlike domain adaptation that only requires adaptation in the input space, this new problem requires adaptation in both the input and output spaces. To address this problem, we proposed AITL, an Adversarial Inductive Transfer Learning method which, to the best of our knowledge, is the first method that addresses the discrepancies in both the input and output spaces. AITL uses a feature extractor to learn features for target and source samples. Then, to address the discrepancy in the output space, AITL utilizes these features as input of a multi-task subnetwork that makes predictions for the target samples and assign cross-domain labels to the source samples. Finally, to address the input space discrepancy, AITL employs global and class-wise discriminators for learning domain-invariant features. In our motivating application, pharmacogenomics, AITL adapts the gene expression data obtained from cell lines and patients in the input space, and also adapts different measures of the drug response between cell lines and patients in the output space. In addition, AITL can also be applied to other applications such as rare cancer diagnosis or predicting long-term clinical labels for slow progressing cancers. We evaluated AITL on four different drugs and compared it against state-of-the-art baselines from three categories in terms of AUROC and AUPR. The empirical results indicated that AITL achieved a significantly better performance compared to the baselines showing the benefits of addressing the discrepancies in both the input and output spaces. We conclude that AITL may be beneficial in pharmacogenomics, a crucial task in precision oncology.

For future research directions, we believe that the TCGA dataset consisting of gene expression data of more than 12,000 patients (without drug response outcome) can be incorporated in an unsupervised transfer learning setting to learn better domain-invariant features between cell lines and cancer patients. In addition, we did not explore the impact of the chemical structures of the studied drugs in the prediction performance. We believe incorporating this input with transfer learning in the genomic level can lead to a better performance. Currently, AITL borrows information between the input domains indirectly via its multi-task subnetwork and assignment of cross-domain labels. An interesting future direction can be to exchange this information between domains in a more explicit way. Moreover, we also did not perform theoretical analysis on this new problem of transfer learning and we leave it for future work. Finally, we did not distinguish between different losses in the multi-task subnetwork, however, in reality patients are more important than cell lines, and considering a higher weight for the corresponding loss in the cost function can improve the prediction performance.

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

# A APPENDIX

## A.1 SUPPLEMENTARY TABLES

Table A1: Definition of biological terminologies

| Term | Definition |
|---|---|
| Cell lines | Human cells that have been immortalized to grow continuously in the laboratory. |
| Patient-Derived Xenografts (PDX) | Tumor tissue taken from a patient and implanted into mice to mimic the microenvironment around the tumor. |
| Chemotherapy drugs | A type of treatment that stops cancer cells' growth by killing them or stopping them from dividing. |
| Targeted drugs | A type of treatment that is designed for a specific type(s) of cancer cells with minor effect on the other cell types. |

Table A2: Characteristics of the datasets

| Dataset | Resource | Drug | Usage | Sample Size |
|---|---|---|---|---|
| GSE55145 (Amin et al., 2014) | clinical trial | Bortezomib | target | 67 |
| GSE9782-GPL96 (Mulligan et al., 2007) | clinical trial | Bortezomib | target | 169 |
| GDSC (Iorio et al., 2016) | cell line | Bortezomib | source | 391 |
| GSE18864 (Silver et al., 2016) | clinical trial | Cisplatin | target | 24 |
| GSE23554 (Marchion et al., 2011) | clinical trial | Cisplatin | target | 28 |
| TCGA (Ding et al., 2016) | patient | Cisplatin | target | 66 |
| GDSC (Iorio et al., 2016) | cell line | Cisplatin | source | 829 |
| GSE25065 (Hatzis et al., 2011) | clinical trial | Docetaxel | target | 49 |
| GSE28796 (Lehmann et al., 2011) | clinical trial | Docetaxel | target | 12 |
| GSE6434 (Chang et al., 2005) | clinical trial | Docetaxel | target | 24 |
| TCGA (Ding et al., 2016) | patient | Docetaxel | target | 16 |
| GDSC (Iorio et al., 2016) | cell line | Docetaxel | source | 829 |
| GSE15622 (Ahmed et al., 2007) | clinical trial | Paclitaxel | target | 20 |
| GSE22513 (Bauer et al., 2010) | clinical trial | Paclitaxel | target | 14 |
| GSE25065 (Hatzis et al., 2011) | clinical trial | Paclitaxel | target | 84 |
| PDX (Gao et al., 2015) | animal (mouse) | Paclitaxel | target | 43 |
| TCGA (Ding et al., 2016) | patient | Paclitaxel | target | 35 |
| GDSC (Iorio et al., 2016) | cell line | Paclitaxel | source | 389 |

Table A3: Performance of AITL and its variants in terms of prediction AUPR

| Method/Drug | Bortezomib | Cisplatin | Docetaxel | Paclitaxel |
|---|---|---|---|---|
| AITL$-AD$ | 0.72±0.04 | 0.85±0.06 | 0.74±0.02 | 0.63±0.02 |
| AITL$-D_G$ | 0.70±0.07 | 0.83±0.06 | 0.76±0.009 | **0.65±0.03** |
| AITL$-DC$ | 0.70±0.05 | 0.81±0.08 | 0.74±0.04 | 0.63±0.02 |
| AITL | **0.76±0.02** | **0.89±0.03** | **0.78±0.007** | 0.63±0.04 |

## A.2 SELECTED HYPER-PARAMETERS

Selected hyper-parameters for MOLI (Sharifi-Noghabi et al., 2019b):

| Drug | Selected hyper-parameters |
|------|---------------------------|
| Bortezomib | 64 (number of nodes in the hidden layer), 1.5 (margin for the triplet loss), 0.0001 (encoder subnetwork learning rate), 40 (epochs), 0.7 and 0.3 (encoder and classifier dropout rates), 0.01 (weight decay), 0.5 (classifier learning rate), 0.2 (regularization coefficient), 36 (batch size). |
| Cisplatin | 64 (number of nodes in the hidden layer), 0.5 (margin for the triplet loss), 0.005 (encoder subnetwork learning rate), 40 (epochs), 0.5 and 0.5 (encoder and classifier dropout rates), 0.001 (weight decay), 0.001 (classifier learning rate), 0.2 (regularization coefficient), 64 (batch size). |
| Docetaxel | 128 (number of nodes in the hidden layer), 1 (margin for the triplet loss), 0.05 (encoder subnetwork learning rate), 25 (epochs), 0.6 and 0.5 (encoder and classifier dropout rates), 0.001 (weight decay), 0.001 (classifier learning rate), 0.1 (regularization coefficient), 36 (batch size). |
| Paclitaxel | 64 (number of nodes in the hidden layer), 1 (margin for the triplet loss), 0.0001 (encoder subnetwork learning rate), 15 (epochs), 0.5 and 0.5 (encoder and classifier dropout rates), 0.0001 (weight decay), 0.001 (classifier learning rate), 0.3 (regularization coefficient), 14 (batch size). |

Selected hyper-parameters for ADDA (Tzeng et al., 2017):

| Drug | Selected hyper-parameters |
|------|---------------------------|
| Bortezomib | 256 (number of nodes in the hidden layer of the feature extractor trained on the source samples, feature extractor of the target samples, and also the input layer of the classifier trained on the source samples), 64 (number of nodes in the hidden layer of the discriminator), 0.01 (learning rate), 20 (epochs), 0.3 and 0.7 (dropout rates for target samples feature extractor and the discriminator, respectively), no weight decay, 16 and 16 (batch size for source and target domains, respectively). |
| Cisplatin | 256 (number of nodes in the hidden layer of the feature extractor trained on the source samples, feature extractor of the target samples, and also the input layer of the classifier trained on the source samples), 64 (number of nodes in the hidden layer of the discriminator), 0.005 (learning rate), 20 (epochs), 0.3 and 0.6 (dropout rates for target samples feature extractor and the discriminator, respectively), no weight decay, 8 and 16 (batch size for source and target domains, respectively). |
| Docetaxel | 1024 (number of nodes in the hidden layer of the feature extractor trained on the source samples, feature extractor of the target samples, and also the input layer of the classifier trained on the source samples), 512 (number of nodes in the hidden layer of the discriminator), 5e-5 (learning rate), 15 (epochs), 0.3 and 0.5 (dropout rates for target samples feature extractor and the discriminator, respectively), 0.005 (weight decay), 16 and 32 (batch size for source and target domains, respectively). |
| Paclitaxel | NA. |

Selected hyper-parameters for ProtoNet (Snell et al., 2017):

| Drug | Selected hyper-parameters |
| --- | --- |
| Bortezomib | 16 (number of nodes in the hidden layer), 5e-5 and 0.5 (learning rates for training on source and target domains), 15 (number of epochs for the source and target domains), 0.7 (dropout rate for the source and target domains), 2 and 8 (number of support and query), 100 (number of episodes). |
| Cisplatin | 256 (number of nodes in the hidden layer), 0.0005 and 0.5 (learning rates for training on source and target domains), 15 and 10 (number of epochs for the source and target domains), 0.3 and 0.4 (dropout rate for the source and target domains), 2 and 4 (number of support and query), 100 (number of episodes). |
| Docetaxel | 16 (number of nodes in the hidden layer), 0.0005 and 0.1 (learning rates for training on source and target domains), 10 and 30 (number of epochs for the source and target domains), 0.3 and 0.6 (dropout rate for the source and target domains), 4 and 8 (number of support and query), 100 (number of episodes). |
| Paclitaxel | NA. |

Selected hyper-parameters for AITL:

| Drug | Selected hyper-parameters |
| --- | --- |
| Bortezomib | 1024 and 1024 (number of nodes in the hidden layers of the feature extractor), 0.0005 (learning rate), 0.2 and 0.4 (regularization for global and class-wise discriminators), 16 and 16 (mini-batch size for the source and target domains), 0.4 (dropout rate), 10 (epoch). |
| Cisplatin | 512 and 16 (number of nodes in the hidden layers of the feature extractor), 0.05 (learning rate), 0.3 and 0.3 (regularization for global and class-wise discriminators), 32 and 8 (mini-batch size for the source and target domains), 0.15 (dropout rate), 25 (epoch). |
| Docetaxel | 512 and 256 (number of nodes in the hidden layers of the feature extractor), 5e-5 (learning rate), 0.1 and 0.8 (regularization for global and class-wise discriminators), 16 and 32 (mini-batch size for the source and target domains), 0.4 (dropout rate), 20 (epoch). |
| Paclitaxel | 1024 and 1024 (number of nodes in the hidden layers of the feature extractor), 0.0001 (learning rate), 0.9 and 0.3 (regularization for global and class-wise discriminators), 32 and 32 (mini-batch size for the source and target domains), 0.5 (dropout rate), 20 (epoch). |

Selected hyper-parameters for (Chen et al., 2017):

| Drug | Selected hyper-parameters |
|---|---|
| Bortezomib | 128 (number of nodes in the hidden layer of the feature extractor), 32 (number of nodes in the hidden layer of discriminators), 0.0001 (learning rate), 20 (epochs), 0.0001 (weight decay), 0.8, 0.3, 0.2, 0.6, 0.2 (dropout rates in featuer extractor, global discriminator, responder class discriminator, non-responder class discriminator, and classifier, respectively), 0.9 and 0.6 (regularization coefficients for class-wise and global discriminators, respectively), 16 and 64 (batch size for source and target domains, respectively). |
| Cisplatin | 512 (number of nodes in the hidden layer of the feature extractor), 128 (number of nodes in the hidden layer of discriminators), 0.0001 (learning rate), 15 (epochs), 0.0001 (weight decay), 0.3, 0.3, 0.5, 0.8, 0.5 (dropout rates in featuer extractor, global discriminator, responder class discriminator, non-responder class discriminator, and classifier, respectively), 0.4 and 0.7 (regularization coefficients for class-wise and global discriminators, respectively), 8 and 32 (batch size for source and target domains, respectively). |
| Docetaxel | 128 (number of nodes in the hidden layer of the feature extractor), 64 (number of nodes in the hidden layer of discriminators), 0.0005 (learning rate), 5 (epochs), 0.0001 (weight decay), 0.6, 0.4, 0.3, 0.7, 0.4 (dropout rates in featuer extractor, global discriminator, responder class discriminator, non-responder class discriminator, and classifier, respectively), 1 and 0.4 (regularization coefficients for class-wise and global discriminators, respectively), 8 and 32 (batch size for source and target domains, respectively). |
| Paclitaxel | 512 (number of nodes in the hidden layer of the feature extractor), 16 (number of nodes in the hidden layer of discriminators), 0.0005 (learning rate), 10 (epochs), 0.1 (weight decay), 0.6, 0.8, 0.8, 0.7, 0.3 (dropout rates in featuer extractor, global discriminator, responder class discriminator, non-responder class discriminator, and classifier, respectively), 1 and 0.8 (regularization coefficients for class-wise and global discriminators, respectively), 64 and 16 (batch size for source and target domains, respectively). |

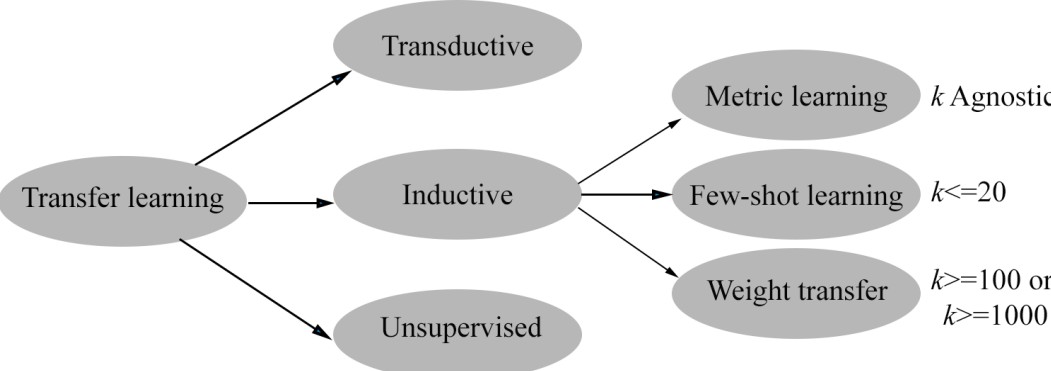

Figure A1: three approaches to inductive transfer learning with respect to the number of samples required for each class in the target domain, denoted as $k$

