# OpenReview forum: "Adversarial Inductive Transfer Learning with input and output space adaptation"
_ICLR.cc/2020/Conference — Reject_

### Official Review · AnonReviewer3 · 2019-10-22
**Official Blind Review #3**

**Rating:** 3

**Review:**

This paper proposes a novel Transfer Learning Method, namely Adversarial Inductive Transfer Learning (AITL), which adapts not only the input space but also the output space. The idea is interesting.

However, the proposed method is not convincing from either theorical analysis or experimental results. No theorical analysis is given on the output space adaption, which is a nontrivial adaption between a discrete probability measure and a continuous probability measure. Furthermore, the method is verified on just one pharmacogenomics case. It is not convincing that the method can be used in another application.


**Experience Assessment:**

I have read many papers in this area.

**Review Assessment: Checking Correctness Of Derivations And Theory:**

I assessed the sensibility of the derivations and theory.

**Review Assessment: Checking Correctness Of Experiments:**

I carefully checked the experiments.

**Review Assessment: Thoroughness In Paper Reading:**

I read the paper at least twice and used my best judgement in assessing the paper.

---

> ### Author Response · Authors · 2019-11-13
> **Response to Blind Review #3**
>
> Thank you very much for your comments and feedback. Although we did not investigate the theoretical aspects of this new problem, we provided enough empirical evidence in our experiments to show the applicability of AITL. We respectfully disagree with the reviewer’s criticism that we studied only one pharmacogenomics case. In fact, we experimented with four different pharmacogenomics datasets. The drugs that we studied have different mechanisms and are being prescribed for different cancers. For example, Docetaxel is a chemotherapy drug mostly known for treating breast cancer, while Bortezomib is a targeted drug mostly used for myeloma. We added a sentence in the experimental section to emphasize this:
>
> “Based on the availability of patient/PDX datasets for a drug, we experimented with four different drugs: Bortezomib, Cisplatin, Docetaxel, and Paclitaxel. It is important to note that these drugs have different mechanisms and are being prescribed for different cancers. For example, Docetaxel is a chemotherapy drug mostly known for treating breast cancer patients (Chang et al., 2005), while Bortezomib is a targeted drug mostly used for multiple myeloma patients (Amin et al., 2014). Therefore, the datasets we have selected cover different types of drugs.“
>
> Moreover, as reviewer 2 mentioned, we are presenting a new problem setting, which is similar to other novel real-world applications that lack well-established benchmark datasets. Our submission does not only propose a method to address this new problem but also provides a list of carefully selected public datasets related to precision oncology that the machine learning community can use to further develop novel methods. We leave a theoretical analysis for future work.  Finally, we would like to note that the authors of our baseline methods adopt a similar experimental design. For example, Tzeng et al., 2017 studied one application (image classification) on two different categories of datasets. Chen et al. 2017 studied one application (image segmentation) on three different datasets. We also studied one application (precision oncology) over four different drugs (datasets).
>
> And we added the following  sentence to the conclusion:
> “Moreover, we also did not perform theoretical analysis on this new problem of transfer learning and we leave it for future work.”

---

### Official Review · AnonReviewer1 · 2019-10-23
**Official Blind Review #1**

**Rating:** 3

**Review:**

The paper proposes an adversarial transfer learning network that can handle the adaptation of both the input space and the output space. The paper is motivated by the application of drug response prediction where the source domain is cell line data and the target domain is patient data. Patient data are usually scarce, hence motivating transferring the knowledge learned from the more widely available cell line data to improve the predictive performance based on the patient data.  The idea of making use of adversarial networks is to learn a representation of the data points that is invariant to whether the data points come from the source domain or the target domain. Experiments on real-world data over four drugs demonstrate the effectiveness of the proposed methods compared to other methods that are not specifically designed for this scenario.

The major utility of the proposed method demonstrated by the paper is its empirical effectiveness on real-world data with four drugs. The reviewer has the following concerns:

1. While the proposed method is capable of handling adaption of the output space between the source domain and the target domain, it makes use of a multi-task subnetwork component, which is a sensible modeling choice but it seems that the idea of leveraging a global discriminator and a class-wise discriminator has been exploited in previous works, as pointed out by the authors in Section 4.2. Therefore, the reviewer is concerned about the modeling contribution made in this paper is somewhat incremental given existing literature.

2. The multi-task subnetwork component itself also seems to be a straightforward and naive way to deal with the different tasks addressed in the source domain and the target domain. For one thing, only the data of the source domain are used to optimize the loss in the source domain and only the data of the target domain are used to optimize the loss in the target domain. While it is understandable that such a decision is due to the lack of ground truth of binary label in the source domain and numeric response in the target domain, ideally, the discriminative process might gain further benefits from borrowing information across the two domains.

3. It is also unclear from the paper how each component of the architecture contributes to the final performance. An ablation study that gets rid of the multi-task subnetwork, global discriminator, and class-wise discriminators (and potential combinations of these components) could help to provide better insights in the importance of each component and determine whether empirically these components do play a role coinciding with the description of the paper.

Other issues/questions:
1. Intuitively, what is the purpose of the shared layer g() in section 3.2.2?
2. In the spirit of the class-wise discriminators, will it be helpful to also add a discriminator based on the value of IC50?
3. Concepts and terminology are not well explained. e.g. what is a cell line? The authors should also provide further descriptions of what do the cell line and patient datasets look like early on in the paper. It was not until section 3 it is clear to the readers that in the dataseets consider in this paper, the source domain and the target domain have the same raw feature representation. Furthermore, in section 2, the authors could consider providing a taxonomy in table format to better explain different types of transfer learning and the three approaches to inductive transfer learning.


**Experience Assessment:**

I have read many papers in this area.

**Review Assessment: Checking Correctness Of Derivations And Theory:**

N/A

**Review Assessment: Checking Correctness Of Experiments:**

I assessed the sensibility of the experiments.

**Review Assessment: Thoroughness In Paper Reading:**

I read the paper at least twice and used my best judgement in assessing the paper.

---

> ### Author Response · Authors · 2019-11-13
> **Response to Blind Review #1**
>
> Q 1. Thank you very much for your comments and constructive feedback. We agree that the AITL method might not be novel per se, however, we argue that it is not incremental and in fact it is quite original. As reviewer 2 mentioned, we presented a new problem and showed its real-world applications. We use precision oncology as the driving application for the proposed problem of transfer learning and suggest two other potential applications in the discussion section.  We would like to argue that AITL combines existing methods in a novel way, which enables AITL to address this problem effectively.
>
> Q 2. We agree with you that direct information exchange between different tasks may be beneficial and may improve the performance. However, we would like to point out that AITL achieves this goal indirectly, jointly training source and target domains via the multi-task subnetwork and assigning the cross-domain labels to the source domain. We added a new sentence to the conclusion to emphasize this.
>
> Q 3. We agree with you that the contribution of each component should be studied separately. We selected our baselines exactly in a way to investigate that. We had baselines without transfer learning to see whether or not transfer learning is adding any value. We had a baseline only with the global discriminator (ADDA) to see if one type of adaptation is enough. We also had a baseline with global and class-wise adaptation (Chen et al. 2017) to study the impact of further regularizing domain-invariance without the multi-task component. Finally, we studied all of these components together in our method (AITL) and showed better performance. Therefore, our experiments address your concern.  Nonetheless, to further strengthen our ablation study, we now performed experiments with three new baselines that remove different components of AITL: 1) AITL-AD which utilizes the feature extractor and the multi-task subnetwork, 2) AITL-DG which utilizes the feature extractor, the multi-task subnetwork, and the class-wise discriminators, and 3) AITL-DC which utilizes the feature extractor, the multi-task subnetwork, and the global discriminator. We extended table 1 and added table A3 (Appendix) to include these new experimental results and observed that none of these new baselines achieves the same level of performance as AITL using all of its components together; this again indicates the importance of adaptation in both spaces. We added two paragraphs to the experimental design and the results sections of the paper regarding this.
>
> Issue 1. g(.) provides the source and target tasks with the same feature space so that they will be jointly trained on that feature space with fewer parameters than alternative approaches, such as soft sharing, where there is no shared layer. Instead, in soft sharing,  separate parameters of the source and target networks are regularized to be close to each other.
>
> Issue 2. IC50 is a continuous measure, therefore it is not possible to have a discriminator for it with its raw (continuous) values. It is possible to discretize it, as we had to do for those baselines that cannot work with continuous values. However, discretizing the IC50 values with thresholding causes loss of information. The Waterfall method that we employed for those baselines is a commonly used approach that first generates a waterfall distribution for the drug sensitivity measurements (e.g., IC50) by sorting, and then computes a threshold that separates the data into binary classes based on the Pearson correlation coefficient.
>
> Issue 3.  We added a new table (Table A1 in appendix) for the definition of some of the biological terminologies. and also added the description of the data to the introduction.
>
> Issue 3 (taxonomy). Such a taxonomy is already available in (Pan & Yang, 2009) but we added a new figure to the Appendix (Figure A1) to show three types of inductive transfer learning.

---

> > ### Author Response · Authors · 2019-11-13
> > **Response to Blind Review #1 (added sections to the submission)**
> >
> > Q2.
> > Discussion section:
> > “Currently, AITL borrows information between the input domains indirectly via its multi-task subnetwork and assignment of cross-domain labels. An interesting future direction can be to exchange this information between domains in a more explicit way.”
> >
> > Q3.
> > Experimental design section:
> > “In addition to the experimental comparison against published methods, we also performed an ablation study to investigate the impact of the different AITL components separately. AITL-AD denotes a version of AITL without the adversarial adaptation components, which means the network only contains the multi-task subnetwork. AITL-DG denotes a version of AITL without the global discriminator, which means the network only employs the multi-task subnetwork and class-wise discriminators. AITL-DC denotes a version of AITL without the  class-wise discriminators, which means the network only contains the multi-task subnetwork and the global discriminator.”
> >
> > Result section:
> > “We observed that AITL, using all of its components together, outperforms all the additional baselines omitting some of the components. This indicates the importance of both input and output space adaptation. The only exception was for the drug Paclitaxel, where AITL-DG outperforms AITL. We believe the reason for this is that this drug has the most heterogeneous target domain (see Table A1 in the appendix), and therefore the global discriminator component of AITL causes a minor decrease in the performance.”
> >
> > Issue 3.
> > Introduction section:
> > “In our driving application, the source domain is the gene expression data obtained from the cell lines and the target domain is the gene expression data obtained from patients. Both domains  have the same set of genes (i.e., raw feature representation).”

---

### Official Review · AnonReviewer2 · 2019-10-24
**Official Blind Review #2**

**Rating:** 6

**Review:**

the paper studies transfer learning, which addresses the inconsistencies of the source and target domains in both input and output spaces. usually, we only worry about the inconsistencies in the input domain but here we worry about input and output. the paper proposes adversarial inductive transfer learning which uses adversarial domain adaptation for the input space and multi-task learning for the output space.

the main contribution of the paper is in identifying a new type of problem that may be worth studying. the proposal of the paper is sensible and its potential application to pharmacogenomics seems appealing. the paper shows promising performance of the proposal across datasets.

**Experience Assessment:**

I do not know much about this area.

**Review Assessment: Checking Correctness Of Derivations And Theory:**

I assessed the sensibility of the derivations and theory.

**Review Assessment: Checking Correctness Of Experiments:**

I assessed the sensibility of the experiments.

**Review Assessment: Thoroughness In Paper Reading:**

I read the paper at least twice and used my best judgement in assessing the paper.

---

> ### Author Response · Authors · 2019-11-13
> **Response to Blind Review #2**
>
> Thank you very much for your comments on our paper. We appreciate the fact that you recognize the new problem that we presented in the paper. We added a new sentence to the conclusion section to emphasize this:
>
> “In this paper, we introduced a new problem in transfer learning motivated by applications in pharmacogenomics. Unlike domain adaptation that only requires adaptation in the input space, this new problem requires adaptation in both the input and output spaces. To address this problem, we proposed AITL, an Adversarial Inductive Transfer Learning method which, to the best of our knowledge, is the first method that addresses the discrepancies in both the input and output spaces.”

---

### Decision · Program_Chairs · 2019-12-19

**Decision:**

Reject

**Comment:**

The paper proposes an adversarial inductive transfer learning method that handles distribution changes in both input and output spaces.

While the studied problem is interesting, reviewers have major concerns about the incremental modeling contribution, the lack of comparative study to existing methods and ablation study to disentangling different modules. Overall, the current study is less convincing from either theoretical analysis or experimental results.

Hence I recommend rejection.